# Analysis of Rac/Rop Small GTPase Family Expression in *Santalum album* L. and Their Potential Roles in Drought Stress and Hormone Treatments

**DOI:** 10.3390/life12121980

**Published:** 2022-11-26

**Authors:** Yu Chen, Shengkun Wang, Xiaojing Liu, Dongli Wang, Yunshan Liu, Lipan Hu, Sen Meng

**Affiliations:** 1College of Biology and Food Engineering, Chongqing Three Gorges University, Chongqing 404100, China; 2State Key Laboratory of Tree Genetics and Breeding, Research Institute of Tropical Forestry, Chinese Academy of Forestry, Guangzhou 510520, China

**Keywords:** *Santalum album*, *Rac* gene family, haustorium, tissue-specific expression, drought stress, hormone treatments

## Abstract

Plant-specific Rac/Rop small GTPases, also known as Rop, belong to the Rho subfamily. Rac proteins can be divided into two types according to their C-terminal motifs: Type I Rac proteins have a typical CaaL motif at the C-terminal, whereas type II Rac proteins lack this motif but retain a cysteine-containing element for membrane anchoring. The *Rac* gene family participates in diverse signal transduction events, cytoskeleton morphogenesis, reactive oxygen species (ROS) production and hormone responses in plants as molecular switches. *S. album* is a popular semiparasitic plant that absorbs nutrients from the host plant through the haustoria to meet its own growth and development needs. Because the whole plant has a high use value, due to the high production value of its perfume oils, it is known as the “tree of gold”. Based on the full-length transcriptome data of *S. album*, nine *Rac* gene members were named *SaRac1-9*, and we analyzed their physicochemical properties. Evolutionary analysis showed that SaRac1-7, AtRac1-6, AtRac9 and AtRac11 and OsRac5, OsRacB and OsRacD belong to the typical plant type I Rac/Rop protein, while SaRac8-9, AtRac7, AtRac8, AtRac10 and OsRac1-4 belong to the type II Rac/ROP protein. Tissue-specific expression analysis showed that nine genes were expressed in roots, stems, leaves and haustoria, and *SaRac7*/*8*/*9* expression in stems, haustoria and roots was significantly higher than that in leaves. The expression levels of *SaRac1*, *SaRac4* and *SaRac6* in stems were very low, and the expression levels of *SaRac2* and *SaRac5* in roots and *SaRac2*/*3*/*7* in haustoria were very high, which indicated that these genes were closely related to the formation of *S. album* haustoria. To further analyze the function of *SaRac*, nine *Rac* genes in sandalwood were subjected to drought stress and hormone treatments. These results establish a preliminary foundation for the regulation of growth and development in *S. album* by *SaRac*.

## 1. Introduction

*S. album* is a small semiparasitic tree belonging to the genus *Santalum* in the family *Santalaceae*. This family is composed of 29 genera with approximately 400 species, 19 of which are specific to the *Santalum* genus [1,2,3]. It is distributed in India, Malaysia, Australia and Indonesia and is also cultivated in Guangdong and Taiwan [4]. *S. album* is known as the “tree of gold”, and its heartwood is widely used in high-end craft sculpture and high-end furniture manufacturing [5,6]. The essential oil extracted from its heartwood is mainly used in the cosmetic and pharmaceutical industries due to its special aroma [7,8]. Sandalwood usually yields 3–7% essential oil depending on the region and hemisphere [9]. The value of a sandalwood tree depends on three important characteristics: the volume of heartwood and the concentration and quality of its heartwood oil [10]. *S. album* grows for a long time; generally, it begins to form heartwood after 7 or 8 years of growth in natural environments. Unsustainable exploitation of wild trees, combined with increasing global demand for sandalwood products has threatened native sandalwood populations in some places, such as southern India. Because of the contradiction between the increasing market demands for its heartwood and the shortage of *S. album* heartwood on the market, the shortage of heartwood with its destruction of natural resources has become increasingly complicated [11].

*S. album* is a semiparasitic plant; its roots have a special organ, the haustorium, which, through the haustorium’s contact with the host plant root, absorbs its own water and nutrients [12]. Therefore, the growth and development of haustoria play a vital role in the growth and development of *S. album* and the formation of heartwood. As far as the current scientific progress is concerned, the study of the regulatory genes related to haustoria in *S. album* is still limited.

ROS can regulate plant growth and development, hormone transduction, root hair development and so on. In sandalwood, Rboh is significantly induced by the haustorial inducer DMBQ, and the ROS signal produced by Rboh protein is necessary for the development of sandalwood haustoria [13]. At present, the growth and development of sandalwood is related to the size, quantity and quality of haustoria. According to the available literature, Rac and Rboh can regulate ROS concentrations through interactions and then promote the formation of sandalwood haustoria. In conclusion, the interaction between Rac and Rboh is an indispensable link in the formation of ROS for haustorium development.

Plant-specific Rac small GTPases belong to the plant Rho subfamily [14]. In animals, Rho is divided into several subfamilies, including Rho, Cdc42 and Rac. Rop is also called Rop in plants [15]. Rac is the sole plant subfamily of the conserved Rho family of small GTPases [16,17,18,19]. They are soluble proteins that localize and function at the plasma membrane by way of posttranslational lipid modifications [20,21,22]. Rac proteins can be divided into two types according to their C-terminal motifs: Type I Rac proteins, which have a typical CaaL motif at the C-terminal, and Type II Rac proteins, which have a truncated and functional motif. Whereas type-I Racs probably undergo prenylation, type-II Racs undergo S-acylation but not prenylation [23]. All type-II Racs have an additional exon at the 3′ end of the gene, probably resulting from the insertion of an intron into an ancestral Rac [24,25]. Most if not all plant Racs act at the plasma membrane [26], and they are molecular switches that regulate diverse signaling cascades [27,28]. They are widely involved in plant signal transduction [29,30,31], cytoskeleton morphogenesis, polarized cell growth [32,33], cell morphogenesis, defense, responses and reactive oxygen species (ROS) production [34,35,36]. Taken together, Rac small GTPases act as a simple binary switch capable of receiving a wide variety of inputs and accordingly generating a multitude of specific outputs. Some members of the plant *Rac* gene family have been shown to regulate the growth and development of *S. album* through ROS production. Based on previous studies, nine *Rac* genes were obtained from the transcriptome and genomic data of *S. album* and compared with eleven *Rac* family members in *A. thaliana* and seven *Rac* family members in rice. The bioinformatics data of these sequences and tissue-specific expression were analyzed and these results will provide a basis for further analysis of the related functions of *S. album Rac* genes and its relationship with haustorium formation. Additionally, the expression levels of *SaRac1-9* under drought stress and hormone treatments were studied, which play an important role in the growth and development of sandalwood.

## 2. Materials and Methods

### 2.1. Plant Materials and Treatments

The materials used in the experiment are stored in this laboratory, Institute of Tropical Forestry, Chinese Academy of Forestry Sciences (Guangzhou, 23°11′ N, 113°23′ E). Four tissues were collected: root, stem, leaf and primary haustorium (after collection, they were immediately immersed in liquid nitrogen), each with three biological replicates, and stored in a −80 °C refrigerator. Sandalwood seedlings grew in a greenhouse in the soil for approximately 2 months, until they reached 15 cm. Saplings displaying similar growth were subsequently used for analysis. The soil water was withheld to drought for 0, 3 and 9 d, in which the soil relative water content was reduced from 70% to 32%. The soil relative water content of control plants was maintained at 80%, which represents a suitable soil environment for sandalwood growth. For hormone treatments (ABA, IAA, ethephon), the plants were divided into four zones (ABA, IAA, ethephon, CK), which were treated with different hormones at a concentration of 100μM/L. Three biological replicates were included. After 48 h, samples were collected at the same time and immediately immersed in liquid nitrogen.

### 2.2. Genome-Wide Identification and Sequence Analysis of Rac Genes in S. album

According to the gene annotation information of the sandalwood transcriptome, the coding sequence of *Rac* was obtained, and the DNA sequence and promoter sequence of *Rac* were obtained from *S. album* genome data by homologous alignment. The nucleotide sequence homology of *Rac* in sandalwood was analyzed using TBtools.

The conserved domains of nine SaRac proteins were predicted using NCBI-Conserved Domain Search, and the structure of SaRac was analyzed using TBtools. Novel putative motifs were explored using the MEME server (https://meme-suite.org/meme/, accessed on 13 July 2022). By selecting a motif of full length, eight conserved motifs located in the SaRac domain were identified. To analyze the cis-regulatory element in the putative *SaRac* promoter, we performed a cis-regulatory element analysis of the 2000 BP promoter sequence of the nine *SaRac* genes using TBtools and the Plant CARE website (http://bioinformatics.psb.ugent.be/webtools/plantcare/html/, accessed on 1 August 2022), and their functions and numbers were predicted.

### 2.3. Analysis of Physicochemical Properties of SaRac

The basic physical and chemical properties of the protein, including molecular weight (MW), isoelectric point (pI) hydrophilicity and number of amino acids, were analyzed using ProtParam online software (https://web.expasy.org/protparam/, accessed on 25 August 2022). Protein subcellular localization was predicted by using the WoLF PSORT website (https://wolfpsort.hgc.jp/, accessed on 27 August 2022).

### 2.4. Phylogenetic Analysis

A multiple sequence alignment of 27 Rac proteins from *S. album* and other species, including *A. thaliana* and rice, was performed. The 11 Rac members of *A. thaliana* were obtained from TAIR (https://www.arabidopsis.org/, accessed on 12 July 2022), and the 7 Rac members of rice were obtained from NCBI (https://www.ncbi.nlm.nih.gov/, accessed on 12 July 2022) using the MUSCLE method. A phylogenetic tree was constructed by using the NJ method implemented in MEGA-X. The parameters for tree construction were as follows. Phylogenetic analysis parameters: bootstrap (1000 replicates); gaps/missing data: pairwise deletion; model: Dayhoff model; pattern among lineages: same (homogeneous) and rates among sites: uniform rates. Finally, the phylogenetic tree was constructed.

### 2.5. Chromosome Location and Gene Structure Analysis

Positional information and gene structures of *SaRac* genes on chromosomes of *S. album* were obtained from the gene annotation information of the *S. album* transcriptome. The chromosomal locations were displayed with TBtools (github.com/CJ-Chen/TBtools, accessed on 30 August 2022). The numbers and organization of introns, exons and gene structures were drawn and displayed using TBtools.

### 2.6. Collinearity of the SaRac Gene

The chromosomal locations of *SaRac* genes were obtained by TBtools software. Using TBtools software, the synteny relationships of orthologous *Rac* genes among *S. album*, *A. thaliana* and rice were evaluated. The parameters for collinearity of the *SaRac* gene were as follows: the genome file was used to construct the chromosome skeleton to obtain the gene density file, and then TBtools was used to prepare the collinearity file, extract the location of the *SaRac* gene, highlight the collinearity region of the target gene and finally, visualize it.

### 2.7. Expression Profiles of SaRac Genes in Different Tissues and under Drought and Hormone Treatments

According to the instructions of the plant RNA extraction kit (R6827, Omega Bio-tek, Norcross, GA, USA), total RNA was extracted from *S. album*. The *SaRac1-9* gene sequence was obtained by reverse transcription of cDNA with a reverse transcription kit (DRR037S, Takara, Dalian, China), and quantitative primers were designed according to the whole-field sequence of the *SaRac1-9* gene (Appendix A). Real-time quantitative PCR primers were designed using Primer 3.0 software with *S. album*. Actin was used as an internal reference gene. Using cDNA as a template, real-time quantitative PCR was performed according to the instructions of the SYBR Green Premix Ex Taq II Kit (Qiagen, Dusseldorf, Germany). The qPCR mixture were 0.5 μL of primers, 1 μL of cDNA, 10 μL of 2 × Mix and 8 μL of dd H_2_O. The reaction procedure was 95 °C for 10 s, 60 °C for 10 s and 72 °C for 20 s, and 40 cycles were performed. Three biological replicates and 3 technical replicates were carried out. SPSS 25 software was used to analyze the significance of the data, and OriginPro 2019b software was used to draw the graph.

## 3. Results

### 3.1. Sequence Analysis of the SaRac Gene Family in S. album

Nine cDNA and DNA sequences of *SaRac* were isolated from the transcriptome and genome of *S. album* and named *SaRac1-9*. The basic physical and chemical properties of the protein were analyzed by ProtParam (Table 1). The nine predicted full-length Rac proteins varied from 196 (SaRac5) to 211 (SaRac7) amino acid residues, and the relative molecular mass ranged from 21.45 (SaRac5) to 23.28 (SaRac7) kDa, with isoelectric points in the range of 9.18–9.62.

### 3.2. Analysis of SaRac Protein Homology

The conserved domains of the nine SaRac proteins were predicted using NCBI-Conserved Domain Search, and the results showed that all nine Racs contained Rop-like domains (Figure 1A) belonging to the *Rop* gene family. The structure of the *SaRac* gene family was analyzed using TBtools (Figure 1B). To investigate the structural diversity of *Racs*, a total of 10 conserved motifs in the *Racs* were captured by the MEME website, and we obtained five conserved motifs located in the Rac domain. We further analyzed the sequence structures of nine *SaRac* genes and aligned them to eleven *Rac* members in *A. thaliana* [37] and seven *Rac* members in rice [38] (Figure 1C,D). It is worth noting that the type and distribution of the C-terminal domains of most *Racs* are similar. The results showed that the nine SaRac proteins all had five conserved motifs, which were the same as the Rac proteins in *Oryza sativa* and *A. thaliana*.

### 3.3. Sequence Analysis of the SaRac Gene Family

Novel putative motifs were explored using the MEME server. By selecting a motif of full length, we identified eight conserved motifs located in the *SaRac* domain (Figure 2A,B).

To analyze cis-acting elements in putative *SaRac* promoters, the 2000 BP promoter sequences of nine *SaRac* genes were identified as cis-regulatory elements by TBtools and the Plant CARE website, and their functions and numbers were predicted (Figure 2C). There were differences in the types and numbers of regulatory elements in the promoters of the nine *SaRac* genes, but all of them had multiple hormone response elements and stress response elements. For example, *SaRac1* and *SaRac4* have two hormone response elements, while *SaRac2* and *SaRac3* have three hormone response elements, *SaRac6-8* have four hormone response elements and *SaRac5* and *SaRac9* have five hormone response elements. In addition, *SaRac3*/*6*/*8* have one stress response element, *SaRac4*/*7*/*9* have two stress response elements and *SaRac5* and *SaRac2* have three and four stress response elements, respectively. Among them, *SaRac6* has ten CGTCA motifs, and we can further speculate that this gene may be related to the chemical defense response of *S. album*.

### 3.4. Chromosome Distribution of the SaRac Gene Family

To clarify the distribution of *SaRac* on the chromosomes of *S. album*, we used TBtools software to map the location of *SaRac* family members (Figure 3A). The *SaRac* family members of *S. album* showed irregular distribution on the chromosome and did not form a large number of gene clusters. The graph shows that nine *SaRac* genes are located on chr6, chr7, chr8, chr9 and chr10 of *S. album*, and more than half of them have one *SaRac* gene member on the chromosome. The remaining two chromosomes have four and one *SaRac* gene members. Notably, in *S. album* linkage groups6/7/8/9/10, the eight *SaRac* genes were classified into five segmental duplication events (*SaRac4*/*SaRac1*, *SaRac4*/*SaRac3*, *SaRac3*/*SaRac1*, *SaRac9*/*SaRac1* and *SaRac6*/*SaRac1*) (Figure 3A,B) (Appendix A).

To further infer the phylogenetic relationship between sandalwood, *A. thaliana* and rice, we constructed two syntenic maps of sandalwood with *A. thaliana* and rice. A total of three *SaRac* genes showed syntenic relationships with *Racs* in *O. sativa* (Figure 4A). In *A. thaliana*, there are many *SaRac* genes that are in common with *Racs* (Figure 4B) (Appendix A).

### 3.5. Phylogenetic Comparison of Rac in Different Species

To further evaluate the phylogenetic relationship between SaRac and other plants, 11 and 7 Rac sequences from *A. thaliana* and rice were compared with 9 Rac sequences from *S. album*, and the phylogenetic tree of the whole protein sequence alignment was constructed by MEGA-X. The results showed that the *Rac* genes of these three species can be divided into five subgroups: I, II, III, IV and V (Figure 5). From the available literature, we further obtained the role of *Rac* in *A. thaliana* and rice. For instance, in *A. thaliana, AtRac1*/*6*/*11* are highly expressed in mature pollen and play an important role in pollen tube growth [37]. *AtRac2* overexpression inhibits the growth of root tips. *AtRac3*/*8* inhibit ABA-induced responses, including actin recombination in guard cells, stomatal closure, seed germination, root elongation and gene expression [39,40]. *AtRac4* is a positive regulator of root hair initiation and apical growth [41]. *AtRac5* acts on actin dynamics, polar growth, root hair growth and so on; finally, *AtRac10* participates in the regulation of membrane transport. On the other hand, in rice, *OsRac1* is a resistance and grain size gene [42,43]. *OsRac4*/*5* are negative regulators of blast resistance, and OsRacB is a direct effector of OsRopGEF2/3/6. It is a potential downstream target of OsRopGEF2/3/6/8 and confers salt tolerance, a negative regulator of disease resistance [44]. In addition, we can more accurately estimate the functions likely to be contained in the members of the nine *SaRac* gene families.

### 3.6. Analysis of Rac/Rop Multibase Regions in S. album, A. thaliana and O. sativa

The members of the *Rac* gene family are divided into type I Rac/Rop protein and type II Rac/Rop protein. Type I proteins have a conserved CaaL motif at the C-terminus. However, type II has a truncated, but functionally modified, posttranscriptional motif. Nine SaRac, eleven AtRac and seven OsRac proteins were sequenced, and the results showed that SaRac1-7 and AtRac1-6, AtRac9 and AtRac11 as well as OsRac5, OsRacB and OsRacD belong to the typical type I, while SaRac8-9 and AtRac7, AtRac8 and AtRac10 and OsRac1-4 belong to type II (Figure 6).

The conserved G domains in the N-terminal region of these proteins were GTPase active domains (G1, G3), Mg^2+^ binding domains (G2) and GTP binding sites (G4, G5). The G2 and G3 domains are also called switch I and II loops, and the C21 and C156 positions are the conserved L-cysteine residues of the G domain (Figure 6).

### 3.7. Tissue Specificity of Rac Expression in S. album

We first investigated the tissue-specific expression of nine *Rac* gene family members in roots, stems, leaves and primary haustoria of *S. album* by green fluorescent quantitative PCR with gene-specific primers. Nine genes (*SaRac1-9*) were expressed in roots, stems, leaves and primary haustoria, but there was a difference in their expression levels. The expression of *SaRac7*/*8*/*9* in stems, haustoria and roots was higher than that in leaves, but the expression of *SaRac1*, *SaRac4* and *SaRac6* in stems was lower in leaves. The expression levels of *SaRac2* in stems and primary haustoria were high. It is worth noting that the expression levels of *SaRac2* and *SaRac5* in roots were approximately 6 and 13, respectively, relative to leaves (Figure 7). Therefore, we can further speculate that *SaRac2* and *SaRac5* may have a strong positive correlation with the growth and development of haustoria in sandalwood. In contrast, a high expression of *SaRac2* and *SaRac5* may inhibit the expression of *SaRac2* and *SaRac5* in leaves. The tissue-specific expression of *Rac* showed that different *Rac* members play different roles in different signaling pathways of *S. album*.

### 3.8. Expression of SaRac Genes under Drought and Hormones Treatments

To better understand the function of *SaRac* in response to abiotic stress, nine *SaRac* genes were selected for further analysis. Under drought stress, the expression of more than a third of the genes, including *SaRac1*, *SaRac3*, *SaRac4* and *SaRac7* were increased. The general trend of *SaRac1*, *SaRac3*, *SaRac4* and *SaRac7* expression was first increased and finally decreased at 9 d. However, the expression levels of *SaRac2* and *SaRac9* were downregulated. Only two genes, *SaRac6* and *SaRac8*, showed higher expression under long-term drought treatment than those under control conditions (Figure 8A). 

The expression levels of *SaRac1, SaRac6, SaRac7* and *SaRac8* increased after 48 h of IAA treatments, indicating that these genes were responsive to IAA. On the contrary, the expression levels of *SaRac2*, *SaRac4*, *SaRac5* and *SaRac9* were lower under IAA treatments than those under control conditions. Furthermore, more than half of the genes, including *SaRac1*, *SaRac3*, *SaRac5* and *SaRac6*, showed higher expression under ethephon treatments. It is worth mentioning that the expression of *SaRac5* under ethephon-treated conditions was 8 times higher than that under control conditions. Moreover, ABA treatment significantly induced the expression of *SaRac1*, *SaRac3*, *SaRac7* and *SaRac8*. However, *SaRac2*, *SaRac4* and *SaRac9* expression were decreased in response to both ethephon and ABA treatments.

## 4. Discussion

Rac small GTPases are members of the plant-specific Rho subfamily and are involved in many signaling events, such as defense responses, pollen tube growth, root hair development, reactive oxygen species (ROS) production and phytohormone response, and play a very important role in the abovementioned events [45,46]. Rac protein is a soluble protein that localizes in the plasma membrane and functions through posttranslational lipid modification [23,47,48]. For example, 11 *Rac* genes have been identified in *A. thaliana,* and there are eight type I *Rac* genes: *AtRac1-6*, *AtRac9*, and *AtRac11*. Seven *Rac* gene family members were identified in rice, while seven *Rac* family members were identified in *H. vulgare*, indicating that the copy number of *Rac* genes is not the same in different species. Studies have shown that *AtRac1*/*6*/*11* are highly expressed in mature pollen, and *AtRac4*/*2* are a pair of positive and negative regulators of root hair tips. At present, most of the research on *Rac* genes focuses on medicine and animals, while research on plants focuses on model plants, such as *A. thaliana*, *O. sativa* and *Hordeum vulgare*. Little is known about the mechanisms by which *SaRac* impacts the growth of *S. album*. 

Reactive oxygen species (ROS) produced by NADPH oxidase have been shown to play many important roles in signaling and development in plants, such as in plant defense response, cell death, abiotic stress, stomatal closure, and root hair development [49,50,51,52,53]. In sandalwood, they control the development and formation of haustoria [13]. At present, studies have shown that the interaction between Rac GTPases and the N-terminal extension is ubiquitous and that a substantial part of the N-terminal region of Rboh, including the two EF-hand motifs, is required for the interaction [54]. The interaction between Rac and Rboh also provides further theoretical help for the study of the mechanism of Rac. At the same time, the regulation of Rboh ROS production by Rac provides a theoretical basis for the development of sandalwood haustoria.

Auxin, abscisic acid (ABA) and ethephon play key roles in the development of many plants. These three hormones are often used as the main substances in plant hormone response experiments. In this study, we treated nine *Rac* genes with drought and hormones. Only two genes, *SaRac6* and *SaRac8*, showed higher expression under long-term drought stress than those under control conditions. 

The results of hormone treatments indicated that the expression of *SaRac1*, *SaRac3*, *SaRac7* and *SaRac8* were higher under ABA treatments. Moreover, more than half of the genes, including *SaRac1*, *SaRac3*, *SaRac5* and *SaRac6*, showed higher expression under ethephon treatments. Furthermore, the expression levels of *SaRac1*, *SaRac6*, *SaRac7* and *SaRac8* increased after 48 h of IAA treatments, indicating these genes are responsive to IAA. Previous studies had demonstrated that auxin biosynthesis is essential for haustorium in haustorium formation in the root-parasitic plants [55,56]. Thus, these genes may be involved in the ontogeny of the *S. album* haustorium and further influence the growth and development of sandalwood.

In our study, to better understand the evolution of the *Rac* gene family in sandalwood, the structure, conserved motifs, phylogenetic relationships and collinearity of *SaRac* genes were characterized. One conserved motif was located in the Rac domain, suggesting that the Rac domain is conserved among *A. thaliana*, rice and sandalwood. Most of the *SaRac* genes exhibited similar numbers of exons. Phylogenetic analysis revealed that SaRac, AtRac and OsRac proteins could be classified into five subgroups. Groups IV and V contain more Racs that come from all three species, and groups I and II also contain Racs from all three species. Moreover, group III contains one Rac in *A. thaliana*. For instance, SaRac5 to SaRac9 in group IV and SaRac4 to AtRac1 in group V could have expansive functions surrounded by AtRac, SaRac or OsRac proteins with different functions.

Additionally, synteny maps between two representative species and sandalwood were constructed to better understand the phylogenetic relationships. More than 10 pairs were detected in *A. thaliana,* and three pairs were detected in rice, indicating a strong homologous relationship between sandalwood and *A. thaliana*, and a weak homologous relationship between sandalwood and rice.

Gene duplication is a major mechanism underlying the evolution of novel protein functions. We detected five *SaRac* genes that were assigned to segmental duplication events, implying high segmental duplication. These results indicated that some *SaRac* genes were possibly generated by gene duplication.

Most Rac proteins identified to date have been functionally characterized in *A. thaliana* and rice, and their roles include the regulation of root hair initiation and apical growth, hormonal responses, stress responses and so on. Among them, the best-described Rac proteins are the members of group IV (Figure 5). These *Rac* genes are involved in root hair formation and disease resistance. In the phylogenetic tree, we can more accurately estimate the functions likely to be contained in the members of the *SaRac* gene family. For instance, we can predict the role of *SaRac* gene by the *AtRac* members of group IV. Similarly, we can make a preliminary prediction of the role of *SaRac* members in groups I/II/V.

Sandalwood is considered one of the most valuable trees in the world. Its heartwood is often used in carving crafts, cosmetics, medicine and other industries, but its main value lies in the essential oils extracted from the heartwood [57]. Therefore, it is important to investigate whether *SaRac* may be related to their accumulation in sandalwood haustorium tissue, thereby affecting the growth and development of the heartwood of sandalwood and, in turn, the quality of heartwood essential oils [58].

In gene expression, promoters play an important role in regulation, through which gene expression can be changed to change the characteristics of plants. Therefore, the study of promoters is a key step in genetic engineering and gene expression research [59]. In addition to the typical core promoter, there are many regulatory elements controlling the functional expression of genes. Analysis of cis-acting elements revealed that most of the *SaRac* promoters contained a large number of elements related to hormones and stress response. The abscisic acid response element (ABRE) was found in the promoters of *SaRac2-9*, and the gibberellin response element (P-box) was found in the promoters of *SaRac1*/*2*/*4*/*5*/*6*/*7*/*8*/*9* However, *SaRac2*/*3*/*5*/*8*/*9* all had TC-rich repeats related to defense and stress responses, suggesting that these *Rac* genes may be involved in disease resistance and stress resistance of *S. album*.

To understand the function of nine *SaRac* members more accurately, based on the functional prediction of gene promoters, in this study, the expression levels of nine genes at four sites were detected by green fluorescent quantitative PCR. The results showed that, compared with the expression levels of leaves in each gene, the expression levels of *SaRac1*, *SaRac4* and *SaRac6* in stems were very low, while the expression levels of *SaRac2* in stems and primary haustoria were very high, and the expression levels of *SaRac2* and *SaRac5* in roots were approximately 6 and 13, respectively. In general, the level of protein expression in specific tissues is closely related to its function. The high expression of *SaRac2* and *SaRac5* in roots and *SaRac2*/*3*/*7* in haustoria may indicate that these genes are closely related to the formation of haustoria in *S. album*.

## 5. Conclusions

Because of the lack of research on the molecular mechanism of growth and development in *S. album* at present, in this study, the biological information and expression pattern of 9 *Rac* genes in *S. album* were analyzed (Figure 9). These results lay the physical and chemical foundation for further studies of the *Rac* family genes involved in the growth and development of *S. album* and regulation its functions. The perspectives of research on the semiparasitic sandalwood will develop towards the Rac-dependent generation of ROS in promoting haustorium development, which more effectively provides a data base for the growth mechanism of sandalwood.

## Figures and Tables

**Figure 1 life-12-01980-f001:**
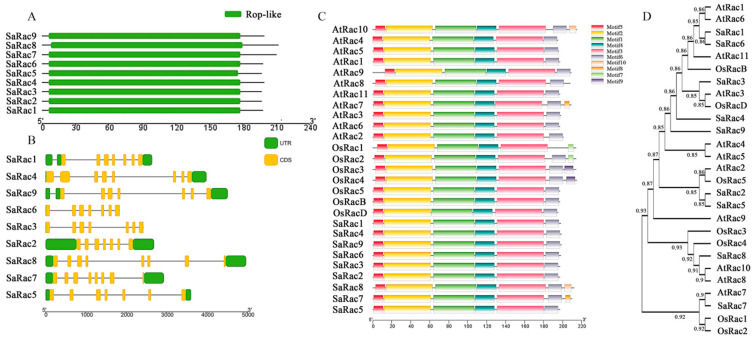
The gene structures and conserved motifs of Rac family members in *S. album* based on evolutionary relationships. (**A**) Rop-like domains of SaRac proteins. (**B**) The exon-intron structure of SaRac proteins. (**C**) Conserved motifs of SaRac proteins. (**D**) Tree of evolutionary relatives: rice, *A. thaliana* and sandalwood.

**Figure 2 life-12-01980-f002:**
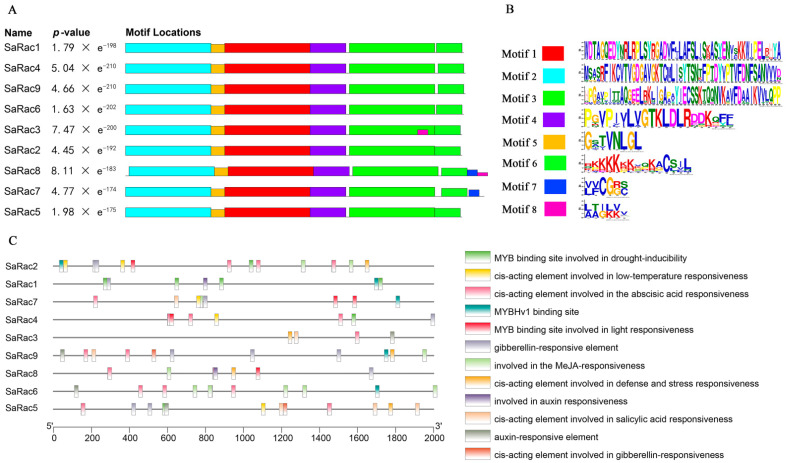
Sequence analysis of the *SaRac* gene family. (**A**) Motif locations of SaRac. (**B**) Discovered motifs. (**C**) Cis-acting Elements in Putative *SaRac* Promoters.

**Figure 3 life-12-01980-f003:**
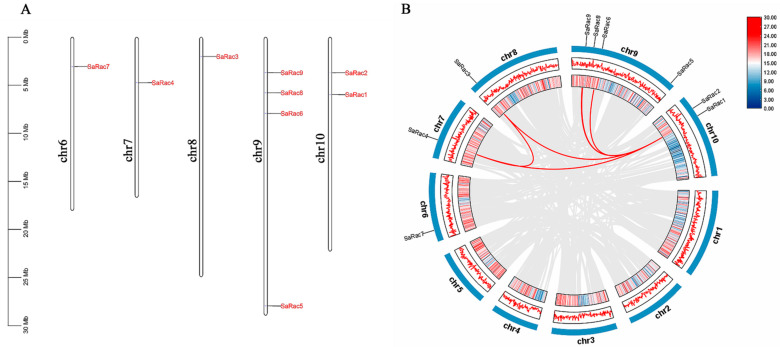
Interchromosomal relationships of *SaRac* genes. (**A**) The location of *SaRac* genes on the *S. album* chromosome. (**B**) Interchromosomal relationships of *Rac* genes in *S. album*. Gray lines indicate all synteny blocks in the *S. album* genome, and red lines indicate segmental duplication events.

**Figure 4 life-12-01980-f004:**
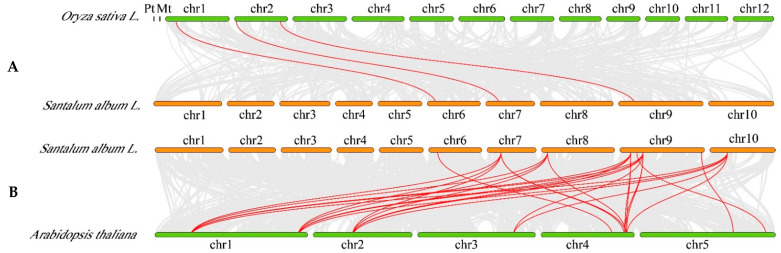
Synteny analysis of *Rac* genes between sandalwood and two representative plant species. (**A**,**B**) Gray lines in the background indicate the collinear blocks within sandalwood and other plant genomes, while the red lines highlight the syntenic *Rac* gene pairs.

**Figure 5 life-12-01980-f005:**
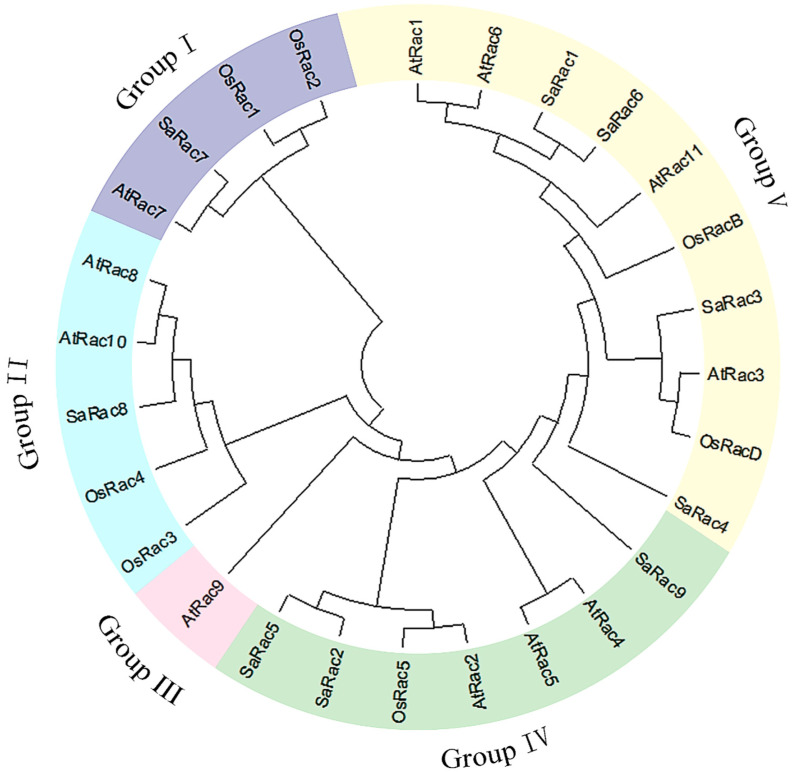
Phylogenetic analysis of Racs from three species (*A. thaliana*, *O. sativa*, *S. album*). Full-length polypeptide sequences were used to make the interspecific phylogenetic tree.

**Figure 6 life-12-01980-f006:**
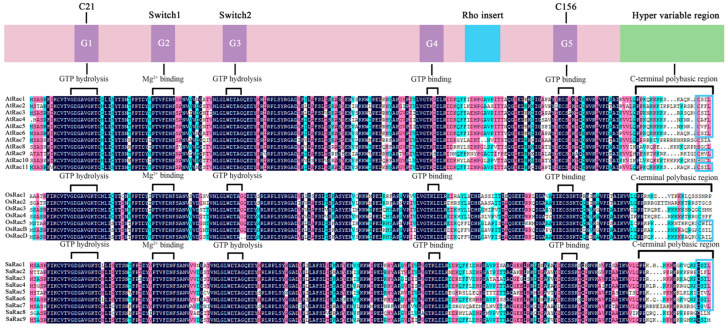
Protein sequence multialignment and domain structure of Racs from *S. album*, *O. sativa* and *A. thaliana*. Conserved motifs are highlighted by blue boxes.

**Figure 7 life-12-01980-f007:**
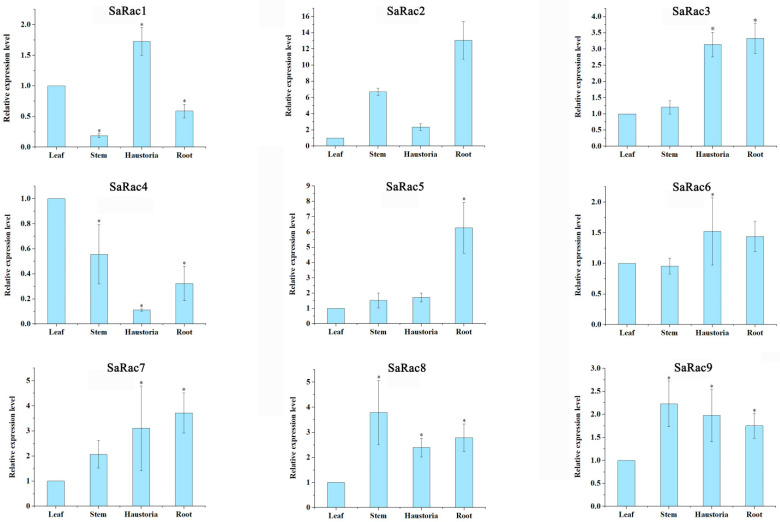
Expression profiles of *SaRac* genes in *S. album* across different organs. The expression level of *SaRac* genes in *S. album* in four organs (leaf, stem, haustoria, root). The relative expression level was calculated by setting the expression value of *SaRac* genes in the leaves of *S. album* at 1. * indicates significant difference by *t* test at *p* < 0.05.

**Figure 8 life-12-01980-f008:**
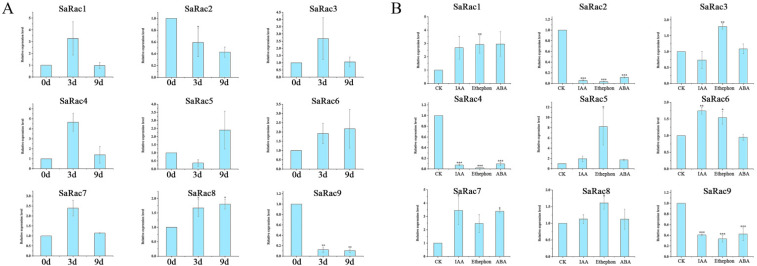
Expression of *SaRac* genes under drought and hormone treatments. (**A**) Expression of *SaRac* genes under drought stress. (**B**) Expression of *SaRac* genes in response to hormones. * indicates significant difference by *t* test at *p* < 0.05; ** indicates significant difference by *t* test at *p* < 0.01; *** indicates significant difference by *t* test at *p* < 0.001.

**Figure 9 life-12-01980-f009:**
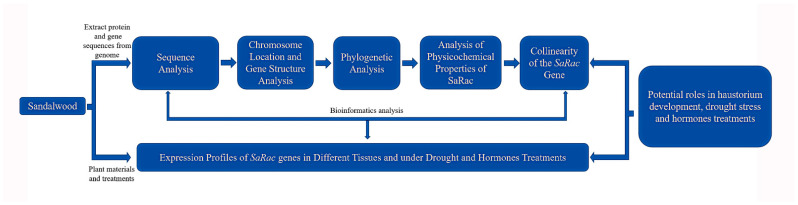
Framework figure. Protein and gene sequences of *SaRacs* were obtained from genome. Bioinformatic analyses were conducted, and the expression profiles of *SaRac* genes in different tissues and under drought and hormones treatments were obtained. These results established a preliminary foundation for the functional study of *SaRac* genes.

**Table 1 life-12-01980-t001:** Chemical properties of proteins in the *SaRac* gene family.

Gene Name	Name	Number of Amino Acids	Molecular Weight (kDa)	PI	Instability Index	Total Number of Negatively Charged Residues (Asp + Glu)	Total Number of Positively Charged Residues (Arg + Lys)	Grand Average of Hydropathicity (GRAVY)	In Silico Prediction WOLF PSORT
Sal10G07090.1	SaRac1	197	21.58	9.20	38.58	18	25	−0.075	plas
Sal10G04200.1	SaRac2	198	21.83	9.43	34.09	16	27	−0.129	chlo
Sal8G02490.1	SaRac3	198	21.81	9.38	40.23	18	25	−0.042	chlo
Sal7G05910.1	SaRac4	197	21.55	9.32	41.48	18	27	−0.106	chlo
Sal9G31620.1	SaRac5	196	21.45	9.25	39.09	16	28	−0.047	chlo
Sal9G09920.1	SaRac6	196	21.74	9.62	38.96	17	25	−0.083	chlo
Sal6G04230.1	SaRac7	211	23.28	9.18	44.44	19	28	−0.069	chlo
Sal9G07020.1	SaRac8	209	22.94	9.27	36.73	19	27	−0.141	plas
Sal9G04490.1	SaRac9	196	21.63	9.55	40.28	18	27	−0.120	cyto

## Data Availability

Not applicable.

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
