# Peer review of "Analysis of Rac/Rop Small GTPase Family Expression in Santalum album L. and Their Potential Roles in Drought Stress and Hormone Treatments"

_life, 2022, doi:10.3390/life12121980_

Round 1
Reviewer 1 Report
· In this study, SaRacs genes were identified using available transcriptome data, and the authors used a computational approach. The data presented in the manuscript was general and preliminary until the relevant data had been dug deeper and further supplemented. I believe the analyses and text should be carefully revised before being published.
· RAC is a plant-specific small GTPases and plays an irreplaceable role in the tissue dynamics of the cytoskeleton, vesicle transport, and hormone signal transmission in plants. So, I suggest checking the gene expression following different abiotic stresses and phytohormones treatments.
· Various cis-acting elements either induce or inhibit gene expression in response to different abiotic stress signals. So I suggest predicting the cis-acting elements in promoters of all the identified SaRacs genes. Also, validate the genes binding affinity with promotor by subcellular localization and yeast one-hybrid (Y1H) assays.
· Add the statistical significance information in figure 7.
Author Response
Reviewer #1:
Q1: RAC is a plant-specific small GTPases and plays an irreplaceable role in the tissue dynamics of the cytoskeleton, vesicle transport, and hormone signal transmission in plants. So, I suggest checking the gene expression following different abiotic stresses and phytohormones treatments.
A: Thank you for your insightful advice. In the present study, we aimed to discover key Rac gene which were closely related to the formation of S. album haustoria and thus we focus on tissue-specific expression analysis of those genes. We would check the gene expression following different abiotic stresses and phytohormones treatments in further research.
Q2: Various cis-acting elements either induce or inhibit gene expression in response to different abiotic stress signals. So I suggest predicting the cis-acting elements in promoters of all the identified SaRacs genes. Also, validate the genes binding affinity with promotor by subcellular localization and yeast one-hybrid (Y1H) assays.
A: Thank you for your suggestions and the cis-acting elements in promoters of all the identified SaRacs genes were added, see Figure 2C. The genes binding affinity with promotor by subcellular localization and yeast one-hybrid (Y1H) assays are conducted in our further research.
Q3: Add the statistical significance information in figure 7.
A: The statistical significance information had been added. See line 224.

Reviewer 2 Report
The article entitled " Analysis of Rac/Rop small GTPase family expression in Santalum album L.". This study mainly aimed to analyze the Rac gene family and their functions in Santalum album. The authors have analyzed the SaRac genes sequences and their coding regions, DNA sequences and promoters. Performed the bioinformatic analyses of these sequences to unravels the function of the Rac gene in regulating the growth and development of Sandalwood. Hence, the manuscript needs some changes. Therefore, I recommend this article for publication after incorporating changes given in below.
Authors must concentrate on the formatting, and use of symbols, etc., check the same throughout the manuscript.
Line 13: Santalum album L. The L. should be un -italics check the whole manuscript.
Line 39: SaRac should be in italics.
Simple summary and Abstract need to be little improved/ reframed with findings.
Two more Keywords should be added. Then the word ‘Parasitism’ was there only in keyword and not there in any parts of the manuscript. Authors are advised to check the same.
The objective of the study in introduction last paragraph (lines: 91-97) is not clear. So, reframe the sentences with clear objective.
Line 115: SaRac should be italics.
Line 144: colinearity should be ‘collinearity’
Authors are advised to add Rac protein stability property to the table 1.
Line 95: ‘and the bioinformatics of these sequences was analyzed to provide’ seems syntax error. Revise it.
Framework figure is required. It will be useful to the readers for better understanding of the studied issue.
Gene name should be in italics. check throughout the manuscript. Check the lines 240-248
Authors have mentioned ‘SaRacs’ in some places. It should be SaRac’s. check and rectify the errors.
In figure 4, synteny analysis Arabidopsis and Oryza’s chromosomes should be in order.
Plant scientific names should be full form in first mention, rest all should be abbreviated format. check throughout the manuscript and revise it. Example: line 184, 239, 254, 256, 269, 300, 302.
Conclusion needs to be written more with the conclusive findings of the study add few lines about future perspectives and hypothesize the current study. It will be useful to the research community to design and understand the importance of studied issue.
In Table S1, the primer should write like forward primer (5’- 3’) and reverse primer (5’- 3’) instead of left and right.
In table S3: add the position of the genes (Eg.: 14569874 to 14569994bp)
Author Response
Reviewer #2:
Q1: Santalum album L. The L. should be un -italics check the whole manuscript.
A: Thank you for your careful check. we had double-checked clerical errors and revised the manuscript. See lines 11, 22, 32, 175, 198, ...
Q2: Two more Keywords should be added. Then the word ‘Parasitism' was there only in keyword and not there in any parts of the manuscript. Authors are advised to check the same.
A: The word ‘Parasitism' was removed and two more keyword ‘Haustorium’ and ‘Tissue-specific expression’ had been added. See line 33.
Q3: The objective of the study in introduction last paragraph (lines: 91-97) is not clear. So, reframe the sentences with clear objective.
A: The sentences had been revised. ‘The bioinformatics data of these sequences and tissue-specific expression were analyzed and these results will provide a basis for further analysis of the related functions of S. album Rac genes and its relationship with haustorium formation.’ See lines 75-77.
Q4: colinearity should be‘collinearity’
A: “ colinearity” had been changed to “collinearity”. See line 116.
Q5: Authors are advised to add Rac protein stability property to the table 1.
A: The Rac protein stability was added. See table 1.
Q6: ‘and the bioinformatics of these sequences was analyzed to provide’ seems syntax error. Revise it.
A: The sentences had been revised. See lines 75-77.
Q7: Authors have mentioned ‘SaRacs’ in some places. It should be SaRac’s. check and rectify the errors.
A: This question was revised, see lines 222-224, lines 249-252...
Q8: Conclusion needs to be written more with the conclusive findings of the study add few lines about future perspectives and hypothesize the current study. It will be useful to the research community to design and understand the importance of studied issue.
A: The conclusion had been revised. See lines 292-294.
Q9: In Table S1, the primer should write like forward primer (5’- 3’) and reverse primer (5’- 3’) instead of left and right.
A: The Table 1 had been revised.
Q10: In table S3: add the position of the genes (Eg.: 14569874 to 14569994bp)
A: The position of all genes had been added. See Table S3.

Round 2
Reviewer 1 Report
I suggest checking the gene expression following different abiotic stresses and phytohormones treatments. Also, genes binding affinity with promotor by subcellular localization and yeast one-hybrid (Y1H) assays. I am not convinced by the author's response. At least the authors should test the gene expression following different abiotic stresses and phytohormones treatments to make their study more meaningful.
Author Response
Reviewer #1:
Q1: I suggest checking the gene expression following different abiotic stresses and phytohormones treatments. Also, genes binding affinity with promotor by subcellular localization and yeast one-hybrid (Y1H) assays. I am not convinced by the author's response. At least the authors should test the gene expression following different abiotic stresses and phytohormones treatments to make their study more meaningful.
A: Thank you for your insightful advice. The drought stress and hormone treatment experiments were added. See lines 225-240...

Reviewer 2 Report
Authors are addressed only partial of the comments. Therefore, I recommend this article for publication after incorporating comments given in below.
Framework figure is required. It will be useful to the readers for better understanding of the studied issue.
Gene name should be in italics. check throughout the manuscript. Check the lines 251-252
In figure 4, synteny analysis Arabidopsis and Oryza’s chromosomes should be in order.
Plant scientific names should be full form in first mention, rest all should be abbreviated format. check throughout the manuscript and revise it. Example: lines: 188, 304, 306
Table 1: In silico should be in italics.
Line 100: S. album Rac should be in italics.
Author Response
Reviewer #2:
Q1: Framework figure is required. It will be useful to the readers for better understanding of the studied issue.
A: The framework figure was added. See figure 9.
Q2: Gene name should be in italics. check throughout the manuscript. Check the lines 251-252
A: Thanks for your careful check. The gene name was revised. See lines 189,302
Q3: In figure 4, synteny analysis Arabidopsis and Oryza’s chromosomes should be in order.
A: The figure 4 was revised. See figure 4.
Q4: Plant scientific names should be full form in first mention, rest all should be abbreviated format. check throughout the manuscript and revise it. Example: lines: 188, 304, 306
A: The plant scientific names were revised. See lines 189,198,200...
Q5: Table 1: In silico should be in italics.
A: ‘In silico’ was revised. See table 1.
Q6: Line 100: S. album Rac should be in italics.
A: The ‘S. album Rac’ was revised. See line 70

Round 3
Reviewer 1 Report
The revision is complete and corrected.